LSTMDD: an optimized LSTM-based drift detector for concept drift in dynamic cloud computing

Mehmood Tajwar 1
http://orcid.org/0000-0002-5801-1568 Latif Seemab 1 seemab.latif@seecs.edu.pk
Jamail Nor Shahida Mohd 2
http://orcid.org/0000-0003-3804-997X Malik Asad 1
http://orcid.org/0000-0001-5304-5948 Latif Rabia 2
1 School of Electrical Engineering and Computer Science (SEECS), National University of Sciences and Technology (NUST) , Islamabad , Pakistan
2 Artificial Intelligence and Data Analytics Laboratory, College of Computer and Information Sciences (CCIS), Prince Sultan University , Riyadh , Saudi Arabia
Aleem Muhammad
Electronic publication date: 2024 Jan 31
Publication date: 2024
Volume: 10
Electronic Location ID: e1827
Received 2023 Aug 30; Accepted 2023 Dec 28
Copyright: © 2024 Mehmood et al.
Copyright year: 2024
Copyright holder: Mehmood et al.
License: This is an open access article distributed under the terms of the Creative Commons Attribution License, which permits unrestricted use, distribution, reproduction and adaptation in any medium and for any purpose provided that it is properly attributed. For attribution, the original author(s), title, publication source (PeerJ Computer Science) and either DOI or URL of the article must be cited.
License URL: https://creativecommons.org/licenses/by/4.0/

Keywords: Concept drift, Drift detection, Cloud usage trace, CPU usage, Memory usage, Machine learning

Funding: The Artificial Intelligence and Data Analytics (AIDA) The authors received no funding for this work. The Artificial Intelligence and Data Analytics (AIDA) Lab at Prince Sultan University paid for the APC of this research article. The funders had no role in study design, data collection and analysis, decision to publish, or preparation of the manuscript.

==============================
This study aims to investigate the problem of concept drift in cloud computing and emphasizes the importance of early detection for enabling optimum resource utilization and offering an effective solution. The analysis includes synthetic and real-world cloud datasets, stressing the need for appropriate drift detectors tailored to the cloud domain. A modified version of Long Short-Term Memory (LSTM) called the LSTM Drift Detector (LSTMDD) is proposed and compared with other top drift detection techniques using prediction error as the primary evaluation metric. LSTMDD is optimized to improve performance in detecting anomalies in non-Gaussian distributed cloud environments. The experiments show that LSTMDD outperforms other methods for gradual and sudden drift in the cloud domain. The findings suggest that machine learning techniques such as LSTMDD could be a promising approach to addressing the problem of concept drift in cloud computing, leading to more efficient resource allocation and improved performance.

Introduction

Cloud computing is a paradigm that facilitates the provision of various services to users via the internet (Zaharia et al., 2011; Saraswat & Tripathi, 2020; Jangjou & Sohrabi, 2022). It offers five essential characteristics: on-demand self-service, ubiquitous network access, resource pooling, rapid elasticity (enabling flexible resource allocation based on demand), and measured services (Puthal et al., 2015; Sunyaev & Sunyaev, 2020; Kamanga et al., 2023). Thus, cloud providers face numerous challenges in ensuring high-quality services.

Efficient resource utilization is also a significant challenge that can directly impact energy consumption (Bharany et al., 2022; Rahimikhanghah et al., 2022; Mansouri et al., 2023). Despite the various methodologies available, this field of study is still emerging due to heterogeneous workloads that fluctuate with cloud-supported services. To address this problem, cloud providers are progressively implementing advancements in strategies like virtualization and server consolidation to maximize resource efficiency and minimize energy usage. Cloud usage prediction aims to allocate resources and adjust server capacity more efficiently. By predicting demand, cloud providers can reduce idle or underutilized servers, optimize energy consumption, manage costs, reduce carbon footprints, and enhance service quality. Therefore, adopting green and sustainable computing practices represents the future direction for cloud providers. According to a study conducted by Google, the energy consumption attributed to idle resources within data centers accounts for over 50% of the total energy usage (Bohrer et al., 2002; Barroso & Hölzle, 2007). Notably, energy consumption scales linearly with resource utilization (Fan, Weber & Barroso, 2007; Reiss et al., 2012). Underutilized or zombie servers pose a significant concern for cloud providers regarding their reputation, as they can result in resource inefficiency and a decline in service quality. Furthermore, implementing proactive monitoring systems can aid in identifying and decommissioning zombie and underutilized servers, promoting a more sustainable and dependable cloud infrastructure.

Organizations are increasingly embracing cloud computing because of its ability to provide resources rapidly and its pay-as-you-go pricing model. Consequently, the workload associated with cloud adoption varies across different organizations (Badshah et al., 2023; El-Rashidy et al., 2023). The workload usage traces generate log data rapidly and exhibit high dynamism. In order to comprehend changes in distribution, it is imperative to analyze these usage patterns, commonly known as concept drift (Lu et al., 2018). The accurate prediction of efficient resource usage is of utmost importance; however, the presence of drift poses a significant challenge to the model’s performance. Utilizing a static model is not a viable solution due to the impact of non-stationary traces over an extended duration on the model’s performance. Hence, using an online learning model becomes necessary to address the phenomenon of concept drift (Agrahari & Singh, 2022). Although online learning models offer advantages in adapting to drift in distribution, they also entail certain drawbacks. The stability-plasticity dilemma can be conceptualized as a dynamic balance between two fundamental aspects: stability, which involves the retention and preservation of valuable knowledge, and plasticity, which necessitates the capacity to effectively adapt to new and unfamiliar information (Bayram, Ahmed & Kassler, 2022). It is crucial to maintain a careful equilibrium in actively identifying changes to prevent needless adjustments to the model that can result in higher computing costs while considering domain expertise.

Monitoring & control and analytic & diagnostic are two crucial concept drift domains covered in the article (Žliobaitė, Pechenizkiy & Gama, 2016). Monitoring and control involve overseeing and regulating tasks and sudden changes caused by complex processes and sensor data input. Diagnostics and analytics often use relational data analysis and observe gradual drift; abrupt changes are rare. After reviewing the literature, it can be concluded that cloud usage traces can be classified in both categories. In the publications mentioned earlier, authors (Fehling et al., 2014; Amiri, Mohammad-Khanli & Mirandola, 2018) have examined the patterns of cloud workload and noted that they can be categorized as static, periodic, one-time occurrences, unexpected, and subject to constant change as a result of user routines. Hence, the assertions above indicate sudden and gradual shifts in cloud workload patterns. Cloud behavior can best be observed with cluster usage traces that allow work on real workloads and their operational complexity. Google released two public usage traces in 2014, and 2019 represents the best benchmark for cloud research. In Chen et al. (2014), this dataset was used to design the model mentioned above and is considered a baseline for performing this research. The Kolmogorov-Smirnov (KS) test detects drift in the dataset. A KS statistical test was applied to the Google usage trace to see drift in the distribution with positive results (dos Reis et al., 2016). In Žliobaitė, Pechenizkiy & Gama (2016) the article, they performed statistical analysis on Google usage trace and identified a multi-model and skewed distribution. We conducted a normal probability plot on the CPU and memory resources of Google data, as shown in Fig. 1. It is a non-Gaussian distribution that does not follow the red line. Moreover, the cloud-generated data stream is time-critical for prediction, mainly because of the changing speed of the data (Gonçalves et al., 2014). There are three significant types of drift concerning overtime, termed abrupt, gradual, and reoccurring (Webb et al., 2016; Khamassi et al., 2018). The abrupt changes are sudden drifts, while gradual drifts are over a more extended period. The reoccurring drifts repeat themselves after time intervals. Real drift refers to changes in only the label class of data concerning time. In contrast, changes in the distribution of input features are the result of virtual drift (Webb et al., 2016). Changes in the distribution of usage load concerning time are called sudden and gradual drift. This drift could be related to any resource, while our focus is the virtual drift.

Figure 1 Google CPU & memory normal distribution plot.

The problem statement concerns monitoring and detecting concept drift in cloud resource usage, particularly emphasizing non-Gaussian continuous resource values. Drift detectors based on data distribution, such as (Webb et al., 2017), do not provide a constant quantitative measure of drift, whereas (Yu & Webb, 2019) limits the restriction to the type of distribution. A specialized concept-drift mapping technique is required to address this challenge. The article’s primary motivation is the model’s involvement in the drift detector to gain more information about drift occurrences. This can help create smaller and optimum buffers for cloud data streams. The article focuses on detecting drift using past knowledge of data distribution in continuous data flow. Note that this updated research question emphasizes detecting and adapting to concept drift in the non-Gaussian distribution in dynamic cloud computing environments and proposes using a drift detector to facilitate proactive retraining. We also compare the performance of existing drift detectors for sudden and gradual drift. Finally, we offer a framework for drift detection to handle the dynamic cloud environment. The framework adopts the changes and learns from the data series to facilitate cloud providers by keeping the following points in mind: How can we effectively detect concept drift in non-Gaussian distributed cloud computing environments and manage adaptive retraining of machine learning models to maintain high accuracy? Specifically, we propose incorporating a focus-based LSTM model with class weight to address the class imbalance in the data and applying a genetic algorithm-based hyperparameter tuning approach to optimize model performance. Additionally, the proposed drift detector can monitor changes in the distribution of cloud computing data and trigger retraining based on the drift when necessary.

The key contributions of the proposed framework are listed below: The objective of this study is to discover the optimum drift detector within the context of cloud computing by utilizing usage trace data obtained from Google.

The examination of individual drift detectors within the context of sudden and gradual drift occurrences.

The proposed approach integrates LSTM models, attention mechanisms, and genetic algorithms to tackle the problem of concept drift detection in non-Gaussian cloud environments.

Evaluating the performance of the proposed approach against existing state-of-the-art methods for detecting concept drift in cloud environments. To help establish the proposed approach’s relative effectiveness and highlight areas for further improvement.

The following sections are in this article. In “Literature Review”, the literature review covers different types of existing drift detectors. The materials and method are covered in “Materials and Method”. The dataset subsection explains the type of dataset used. Then, the following subsection proposes an architecture and model for the cloud drift detector. The results and discussion “Conclusion” with evaluation setup, parameter configuration, and results are discussed. The conclusion summarizes the findings and presents future work.

Literature review

The utilization of online learning is increasingly prevalent in various domains due to the vulnerability of systems to concept drift. Multiple factors, such as modifications in user behavior, environmental changes, or alterations in the system itself, can trigger concept drift. The occurrence of concept drift can significantly degrade the performance of machine learning systems. Therefore, detecting and adapting to concept drift is crucial. This literature review explores the types of concept drift, methods for detecting and adjusting to it, challenges faced, and future research directions in this field.

Two primary methodologies can be employed to demonstrate the occurrence of drift in our data. One approach involves using visualization techniques to discern patterns, while another involves applying drift detector algorithms. In the literature, detectors are classified into three categories: Sequential analysis-based statistical, Data distribution-based drift (DDB), and Learner output-based drift detection. Both sequential analysis-based and data distribution tests employ statistical tests to identify the presence of drift. Sequential analysis-based statistics concentrate mainly on analyzing changes in probability within sequential data, while data distribution-based statistics compare the distributions of two distinct time windows. Two contributions can be compared using either parametric or non-parametric tests. Non-parametric methods are advantageous due to their reduced reliance on prior knowledge of the Probability Density Function(PDF) of data, enhancing their applicability in real-world scenarios. However, this characteristic also renders them less responsive to slight variations.

The concept drift detector uses the results of the base learner. Drift detector selection is based on the drift type in the distribution. It can be used with an individual or an ensemble of classifiers. These methods mostly use a base learner to detect the drift. The drift Detection Method (DDM) (Gama et al., 2004), Early Drift Detection Method (EDDM) (Baena-Garcıa et al., 2006), Adaptive Windowing (ADWIN) (Bifet & Gavalda, 2007), Statistical Test of Equal Proportions(STEPD) (Nishida & Yamauchi, 2007), EWMA for Concept Drift Detection (ECDD) (Ross et al., 2012), DDMs Based on Hoeffding’s (HDDMA) (Frías-Blanco et al., 2015), DDMs Based on Hoeffding’s (HDDMW) (Frías-Blanco et al., 2015), SEED Drift Detector (SEED) (Huang et al., 2014), Sequential Drift (SeqDrift) 1 and 2 (Pears, Sakthithasan & Koh, 2014), and Reactive Drift Detection Method (RDDM) (Barros et al., 2017) are drift detection techniques discussed and compared below in Table 1.

Table 1 Drift detectors summary.

Drift detector	Drift type	Categories	
Page-hinkley test (PHT)	Sudden	Sequential analysis	
Cumulative sum (CUSUM)	Sudden	Sequential analysis	
Drift detection method (DDM)	Sudden	Learner output	
Early drift detection method (EDDM)	Gradual	Learner output	
Adaptive windowing (ADWIN)	Gradual	Learner output	
Statistical test of equal proportions (STEPD)	Sudden	Learner output	
EWMA for concept drift detection (ECDD)	Gradual	Learner output	
DDMs based on hoeffding’s (HDDMA)	Sudden	Learner output	
DDMs based on hoeffding’s (HDDMW)	Gradual	Learner output	
SEED	—	Learner output	
SEQ drift 1 and 2	Gradual	Learner output	
Reactive drift detection (method RDDM)	—	Learner output	

Drift detectors are further classified into three main categories based on their working methodology. The earner output-based drift detection method uses the base learner’s output to detect drift. DDM and EDDM are the most basic learner-based drift detections. It has typically been developed considering classification problems only. When concept drift is detected, they do not provide information about the data necessary to adapt the learner to the new environment. The sequential technique can only deal with a single time series and regression.

Page-Hinkley and H-ICI-based CDT is the most basic used in sequential analysis learners. Its concept drifts over a single time series. Data distribution only works with variable correlation but requires unnecessary model adaptation. Data distribution can help both deal with regression and multivariate data. It can also be used to consider the correlation between attributes; thus, it can extract more detail about the drift. It can further help adopt much data with drift Magnitude, frequency, and duration. The Concept Drift Map is from a Data distribution-based drift detector. It can detect concept drifts of variables irrelevant to the learner output and cause unnecessary ML model adaptations. A concept drift map is more like a visualization technique than a detection one. Table 2 summarizes the characteristics of all three categories of the drift detection technique. Most of the latest work has recently been done in data distribution-based drift detection.

Table 2 Drift detectors types.

Category	Multivariate time series	Regression	Information about data adaption	Model adaption	Variables correlation	
Sequential analysis	No	Yes	Low	Low	No	
Learner output	Yes	No	No	Low	No	
Data distribution	Yes	Yes	High	High	Yes	

Drift Detection Method (DDM): It comes under the category of a learner-based drift detector. A drift detector that utilizes a base learner as its fundamental component. The base learner is employed to make predictions. The error rate is computed by using the classification outcomes. The error rate and standard deviation determine the estimation of drift warnings and dirt occurrences. The number of errors exhibits an inverse relationship with the accuracy of the classification. It is most effective in cases of sudden drift. Let si be the value of the standard deviation i at a given time, pi be an error rate that depends on i, pmin be the minimum value of pi, and smin be the minimum value of si. Then the following drift warning Eq. (1) holds:

(1) (pmin+2⋅smin)≤pi+si

Then, the following drift detection Eq. (2) holds:

(2) pmin+(3⋅smin)≤pi+si

Early Drift Detection Method (EDDM): It is also a learner-based drift detector. Calculate the distance error rate based on the classification results. The number of examples between the two classification errors means that the drift’s distance error rate increases in the case of no concept. The drift warning and occurrence are observed based on the error rate and standard deviation. It works best for a slow, gradual drift. Let pi be the distance error rate and s be the standard deviation. EDDM is calculated using the following Eqs. (3) and (4):

(3) (pi+2)(pmax+(2⋅smax))<α, (Driftwarningcondition)

(4) (pi+2si)(pmax+(2⋅smax))<β, (Driftoccurringcondition)

where α is 0.95 and β is 0.9.

Adaptive Windowing (ADWIN): Its name clearly shows that it uses the sliding window concept and compares two distributions for change. It opens the older window when the change is detected. The window length has been increased based on the fact that no changes have occurred. Two sub-windows are selected to measure the difference.

Statistical Test of Equal Proportions (STEPD): It separately calculates the accuracy of all recent instances to compare two windows. It notifies the presence of the drift by checking for a significant decrease in inaccuracies over the period. The Chi-square test and standard deviation percentile are used to calculate the significance level. A value below the significance level will indicate the drift and drift warning concepts.

EWMA for Concept Drift Detection (ECDD): EWMA stands for an exponential moving average. Probability is calculated based on the learner’s accuracy and the estimated time between false positive detection. Two estimates are calculated and compared. Estimate one is calculated with more weights on the recent examples. Estimate two considers equal importance for all data. The two estimates are compared to detect concept drift. The difference between the two estimates is compared with two thresholds. The first threshold is set to alarm the drift warning, and the second is to detect actual drift occurrence.

DDMs based on Hoeffding’s (HDDMA & HDDMW): HDDMA is based on the moving average, and HDDMW is based on the weighted average. It does not need any assumptions about the PDF.

SEED drift detector (SEED): SEED is based on Hoeffding’s inequality of ADWIN to work faster and more efficiently. It compares two sub-windows by comparing the average. In the case of a higher value, the previous sub-window is dropped.

Reactive Drift Detection Method (RDDM):

RDDM is proposed to overcome the problem of DMM and achieve higher global accuracy. It recalculates the drift warning and identification to remove the effect of previous instances.

The majority of research is devoted to deep learning algorithms for resource usage prediction, such as ensembles (Mehmood, Latif & Malik, 2018), time series (Gutterman et al., 2019), LSTM (Bi et al., 2021) and evolution algorithm (Malik et al., 2022). In Shu, Cai & Xiong (2021), Saba et al. (2023), Banerjee et al. (2023), cloud services are optimized using machine learning to improve resource usage. Deep learning approaches also visualize the drift in cloud (Mehmood & Latif, 2022). The most recent detectors in literature are the advancements of existing drift detectors. Later, a concept drift map was established to understand how drift occurs when all qualities are mentioned in Table 2. The drift mechanism’s efficiency depends on its domain and type. In Webb et al. (2017) understanding, a probability-based idea drift map using Naive Bayes covers the above qualities, but the discrete values make it insufficient for the cloud domain. Thus, continuous values must be binned before determining distance. Translation from a constant process to a categorical conclusion could be more efficient. Transcoding destroys vital data. Thus, discretizing the previous dataset may not work for the present distribution. To overcome strong data type assumptions, (Yu & Webb, 2019) an adaptive concept drift map for continuous values may be proposed. These compare the average mean and maximum values of two Gaussian distributions revealed drift. The research mentioned above highlights a research gap, allowing one to detect deviations when dealing with non-Gaussian distributions and continuous variables. A simple drift detector based on simple Gaussian statistical techniques might also not work perfectly for a complicated domain. Thus, a drift detector based on some knowledge can perform well for a high-intensity stream. In deep learning concept drift detection, resilient methods for changing data distributions are becoming more popular. Traditional ensemble learning and sliding window approaches are well-studied. The Meta-cognitive Online Sequential Extreme Learning Machine (Mirza & Lin, 2016) addresses domain knowledge, imbalanced datasets, and delayed versus quick concept transitions. However, approaches like Dynamic Extreme Learning Machine (Xu & Wang, 2017) and OS-ELM (Yang et al., 2019) stress data flexibility and measure the learner performance degradation. Deep learning models that need dynamic real-world environments fuel this trend. The distribution of real-world data will always differ from model training. The (Cai & Li, 2023) binary classifier detects comparable changes in image distribution when detecting out-of-distribution.

LSTM has been widely used for natural language processing tasks (Jang et al., 2020; Jain et al., 2020; Xie et al., 2020; Li, Xu & Shi, 2019; Zhai et al., 2023), but it has the potential as a drift detector for time-series data. Recent studies (Li et al., 2022; Wang, Qi & Liu, 2019; Fields, Hsieh & Chenou, 2019) worked on the drift-sensitive LSTM method. However, drift is the change that should be part of a model and should not be mitigated for better prediction. The LSTM mechanism can learn the temporal patterns of the data and detect changes in the data distribution, which is yet to be explored. Their promising results in other domains show their power in detecting concept drift in time-series datasets.

Materials and method

This section will cover the dataset, proposed model, parameter configuration, and evaluation mechanism. The dataset section will cover the synthetic dataset used, while cloud data will explain cloud usage traces. The proposed model shows how drift detectors can be used in the cloud. In the section on evaluation methods and techniques to compare the results of these drift detectors. In the subsection, the algorithm’s settings are defined in parameter configuration.

Dataset

Dataset selection is based on previous research (Agrahari & Singh, 2022). Synthetic datasets and benchmarked real cloud usage traces are selected. Two datasets for each abrupt and gradual dataset are chosen. Table 3 shows the type of synthetic dataset with drift types.

Table 3 Dataset description.

Dataset	Drift Type	Dataset Type	
Sine (López Lobo, 2020; Pesaranghader, Viktor & Paquet, 2018)	Abrupt	Synthetic	
Stagger (López Lobo, 2020; Pesaranghader, Viktor & Paquet, 2018)	Abrupt	Synthetic	
Hyperplane (Pechenizkiy, 2004)	Gradual	Synthetic	
Mixed (López Lobo, 2020; Pesaranghader, Viktor & Paquet, 2018)	Gradual	Synthetic	
Google cloud usage trace (Google, 2019)	Abrupt and Gradual	Real	

Sine data represents the sine wave in the coordinate system. Real drift in the label class is observed. When drift occurs, the label representation of positive and negative curves is reversed. Stagger data shows drift by mapping the possible combination of three input variable inputs with the output class. Three shapes, colors, and sizes are used as variables, creating a total of 27 different combinations. When a drift occurs, the same combination will map to another class. The hyperplane is data of hyperplane rotation in the dimension. Two datasets, hyperplane and mixed suffering from gradual concept drifts, were chosen (Hu et al., 2014; Webb et al., 2016).

Cloud google dataset

Google provides the Google usage trace dataset for research purposes to enable a deeper understanding of the cloud’s underlying patterns. The trace exhibits a skewed distribution related to non-Gaussian distribution (Žliobaitė, Pechenizkiy & Gama, 2016). Job duration, network traffic caused by other applications running on the same host with similar parameters, and other factors may impact the drift process. The trace includes CPU, memory, and disk utilization as resources, and task mapping is done using job and machine IDs. CPU actual and requested usage is measured in the core count or core seconds/second, and memory and disk space are measured in bytes. The features used in drift detection include start and end times, unmapped and page cache memory usage, maximum memory usage, cycles per instruction (CPI), memory accesses per instruction (MAI), sampling rate, and aggregation type.

The Kolmogorov-Smirnov (KS) statistical test is a non-parametric method used on the Google dataset to detect drift in distribution. This test compares two distributions, and if the resulting p-value is significantly more than 0.05, the null hypothesis that the distributions are the same is accepted (dos Reis et al., 2016). A higher statistic value indicates a rejection of the null hypothesis. For better visualization, two CPU distribution samples, with one showing changes over time in Fig. 2 and the other comparing two windows of CPU distribution in Fig. 3, may be considered. A sudden drift was observed at the end of the plot, with a spike in CPU usage causing a significant change in the distribution.

Figure 2 Exploring variations in the distribution of cloud resource usage.

Figure 3 An in-depth comparative analysis drift in resource utilization across two windows.

Preparing the dataset for a forecasting model involves data pre-processing, which is conducted in three steps: normalization, handling of missing values, and qualitative coordinates. Regarding normalization, CPU, memory, and disk resource usage measurements are normalized based on the most significant capacity machine on the cluster, where the maximum value is represented as one and the lowest as 0 (Hu et al., 2014). Qualitative coordinates are used to transform real continuous values into a limited set of qualitative classes, particularly for measuring resource usage values, which are continuous and need to be categorized into a limited set of classes. Discretization is applied to convert a continuous set of values into a limited set of discrete values, given the difficulty in detecting which actual value (cores or bytes) refers to the normalized values.

Proposed model and architecture

In dynamic cloud environments, dealing with concept drift using an offline model becomes impossible. There are several types of workload patterns: static, periodic, one-time, sudden, unpredictable, and continuously changing. It will lead to gradual and sudden drifts that can occur in the cloud usage workload trace. However, data distribution-based drift detectors can detect drifts and maintain continuous quantitative measurement. They must be capable of providing a detailed description of the nature and form of ridges. Therefore, an online learning algorithm with a drift detector is required to handle different types of substances and update itself with non-stationary data. Using an outdated model can lead to incorrect resource utilization, so a new solution must be able to handle distribution changes optimally. However, a drift detector with a prediction mechanism has yet to be covered in the literature. LSTM models can store short- and long-term information, but most literature on drift detection is based on the current window size and does not utilize long-term details. Therefore, an online learning workload prediction model with a drift detector is needed to evaluate its accuracy from the cloud provider’s perspective of resource allocation and to improve cloud resource utilization. Finally, it is worth noting that cloud usage workload often has a non-Gaussian distribution (Žliobaitė, Pechenizkiy & Gama, 2016). Figure 4 presents the proposed model for drift detection in cloud data infrastructure. The study offers a framework consisting of three phases: Resource Usage Trace Generation Phase, Online Phase, and Offline Phase.

Figure 4 Long short-term memory based drift detector (LSTMDD).

During the Trace Generation Phase, cloud users generate usage traces when requesting and utilizing cloud resources. These requests are then forwarded to the resource allocator to fulfill the client’s needs. The Resource Usage Monitor captures information such as the client’s timestamp, CPU, Memory, job arrival times, and other relevant data. The Resource Usage Trace maintains storage of these usage traces for further use.

The online learning mechanism has two phases, namely the online and offline phases, which include a data distribution analyzer and a drift detector. In the online step, the trace data is analyzed to make predictions and train a new model in case of sudden changes. The distribution analyzer creates the distribution of data chunks available in the buffer based on the window size. The new model is a base learner trained offline on buffered data. More simply, it is responsible for learning about drift window size, while the drift detector raises alarms when drift occurs. Sudden drift occurs within a shorter period compared to gradual drift. Detecting sudden drifts requires smaller window sizes, while gradual drifts require larger window sizes. In the event of a drift, the online mechanism, which consists of a live model accompanied by a data stream, will be replaced by a new model, also referred to as a fine-tuned model. The offline phase means retraining the new model on buffer data available for the online phase later. The threshold is set to detect the occurrence of gradual drift, and the data is collected until the drift is detected.

LSTM is a type of neural network that can store both short-term and long-term information, making it suitable for prediction in non-stationary environments. It comprises three gates—input, forget, and output—each with defined usage. The memory cell of LSTM is shown in Fig. 5, where the cell state ct contains both short and long memory. The input gate controls the amount of information that can pass from the input xt. The forget gate decides the amount of past information retained, controlled by ft. The output state ot decides the amount of information flow from ct to ht. The hidden state ht predicts based on the output, input, and current cell state to make the final prediction. The training set calculates the prediction error and threshold for detecting gradual drift.

Figure 5 Memory cell representation of LSTM.

A new trace can be created based on the classifier prediction results, and a drift detector can be used to make a drift prediction. LSTM is known for overcoming the problem of vanishing gradient. Each cell design of LSTM is to deal with present and past information. In sudden and gradual drift, the main difference between them is time. Thus, LSTM is the best choice in case of gradual drift in time series data; it can maintain information for a more extended period and abrupt drift in a shorter period. The proposed model employs a sliding window approach to detect changes in the data. At each time step, the model trains an LSTM with hyperparameter tuning on a fixed window of historical data. The LSTM model is trained to predict the next value in the time series. The predicted value is compared with the actual value, and the divergence between the two is computed. If the divergence exceeds a threshold, the model is considered to have detected a drift in the data. The threshold is dynamically updated based on the divergence values observed in the historical data. The proposed Attention Mechanism-based LSTM with class weight is defined using the following steps summarized in the Algorithm 1:

Algorithm 1 Attention mechanism-based LSTM with class weight.

    Require: Training data Xtrain and labels Ytrain, testing data Xtest and labels Ytest, batch size b, epochs e, number of LSTM units u, dropout rate d, learning rate α, class weight w, and optimizer o	
    Ensure: Trained model	
1:     Procedure ATTENTIONLAYER ( inputshape,returnsequences=True)	
2:        b← Initialize biases with zeros	
3:        return Weighted sum of input	
4:     end Procedure	
5:     Procedure CREATE_MODEL ( units=u,dropoutrate=d,optimizer=o,learningrate=α)	
6:        LSTM layer with u units and input shape	
7:        Attention layer	
8:        Dropout layer with dropout rate d	
9:        Dense layer with sigmoid activation	
10:       Compile model with binary cross-entropy loss and optimizer o with learning rate α	
11:        return Compiled model	
12:     end Procedure	
13:      model← CREATE_MODEL( units=u,dropoutrate=d,optimizer=Adam,learningrate=α)	
14:      model.fit(Xtrain,Ytrain,batchsize=b,epochs=e,classweight=w)	
15:      return Trained model	

Input data and labels

1. The input data is represented as X={x1,x2,…,xn}, where xi is a sequence of m feature vectors of dimension d, i.e., xi=x1,1,x2,2,...,xn,m, xi,j is a d-dimensional feature vector.

2. Convert the input data and labels to NumPy arrays.

Trained model and evaluation metrics

Define the attention layer as follows: 1. Initialize the layer with the return_sequences parameter set to True.

2. In the build() method, add the weights W and bias b to the layer. These are initialized with normal and zero distributions, respectively.

3. In the call() method, calculate the dot product of the input tensor and W, add b to the result, and apply the tanh activation function. It produces the energy tensor e.

4. Apply the softmax activation function to e along the time dimension ( axis=1) to produce the attention tensor a.

5. Multiply the input tensor x element-wise with the attention tensor a to produce the output tensor output.

6. If return_sequences is True, return output. Otherwise, return the sum of output along the time dimension. The attention layer is used to weigh the importance of the hidden states hi for the classification task. The attention weights ai are calculated using Eqs. (5) and (6) as follows:

(5) eij=v⋅tanh⁡(W1⋅hj+W2⋅ht+b)

(6) ai=softmax(ei)

Define the create_model function as follows

1. Take units, input_shape, dropout_rate, and optimizer as input parameters.

2. Create a Sequential model. The LSTM model takes the input sequence xi and returns a sequence of hidden states hi=h1,1,h2,2,...,hn,m, where each hi,j is a d-dimensional hidden state vector done using the following equations:

(7) Inputgate:it=sigmoid(Wi⋅[xi,j,hi,j−1]+bi)

(8) Forgetgate:ft=sigmoid(Wf⋅[xi,j,hi,j−1]+bf)

(9) Outputgate:ot=sigmoid(Wo⋅[xi,j,hi,j−1]+bo)

(10) Candidatestate:ct=tanh⁡(Wc⋅[xi,j,hi,j−1]+bc)

(11) Cellstate:(ct=it⋅ct~+ft⋅ct−1)

In the given Eqs. (7)–(11), bf, bo, and bc represent the bias terms for the forget gate, output gate, candidate state, and cell state, respectively. Bias terms are constant values added to the weighted sum of inputs in neural networks. The hidden state in Eq. (12) allows the model to adjust the decision boundary and control the overall output of the gate or neuron.

(12) Hiddenstate:hi,j=ot⋅tanh⁡(ct)

3. Add an LSTM layer with units and input_shape parameters and return_sequences set to True.

4. Add the Attention layer defined above. The final output is obtained by taking a weighted sum of the hidden states hi based on the attention weights ai: Given a query vector q and a set of critical vectors K={k1,k2,…,kn}, the attention weights can be computed as follows in Eq. (13):

(13) Attention(q,K)=softmax(QK)

where softmax denotes the softmax function, Q is a matrix formed by replicating the query vector q, and K is a matrix formed by concatenating the key vectors k1,k2,…,kn along the columns. Once the attention weights are obtained, the final output y in Eq. (14) can be computed as the weighted sum of the hidden states hi using these attention weights:

(14) y=∑i=1n(aihi)

where ai represents the attention weight for the ith hidden state hi.

5. Add a Dropout layer with a rate equal to dropout_rate.

6. Add a Dense layer with a single output neuron and sigmoid activation.

7. Compile the model with binary_crossentropy loss function and the optimizer specified by the optimizer parameter. Use accuracy as a metric.

8. Return the compiled model.

Hyperparameter Tuning: The hyperparameters of the LSTM model, such as the number of LSTM layers, the number of neurons per layer, and the learning rate, are tuned using the Genetic algorithm optimization. It is a probabilistic approach that uses a surrogate model to approximate the objective function and selects the hyperparameters that minimize the expected loss. In this case, the objective function is the divergence between the predicted and actual values.

Genetic Algorithm Hyperparameters Tunning The model’s hyperparameters (i.e., number of units, batch size, dropout rate, and optimizer) are tuned using Genetic Algorithm (GA). The procedure is summarised in the Algorithm 2.

Algorithm 2 Genetic algorithm for hyperparameter tuning in LSTM.

1: Initialize a population of candidate hyperparameters randomly	
2: Set the current generation count g=0	
3: while g<NGENERATIONS do	
4: Evaluate the fitness of each candidate hyperparameter	
5:  procedure FITNESSVALUE( inputshape,returnsequences=True)	
6:    Build and train an LSTM model with the specified hyperparameters	
7:    Evaluate the model on the test data	
8:    Calculate evaluation metrics (accuracy, precision, recall, f-score)	
9:    Compute the confusion matrix	
10:   Calculate false positive rate (FPR) and false negative rate (FNR)	
11:       Returns the accuracy value as the fitness value of the individual.	
12:  end procedure	
13:  Select the top-performing candidates as parents using a tournament method	
14:  Generate offspring by combining the hyperparameters of the parents through crossover	
15:  Apply mutation to the offspring by randomly modifying their hyperparameters	
16:  Replace a portion ( PREPLACE) of the population with the offspring	
17: Increment the generation count g=g+1	
18: end while	
19: Train the LSTM using the best-performing hyperparameters	

1. Define the hyperparameters to be tuned as a dictionary called params. This dictionary includes the following keys: units, batch size, dropout rate, and optimizer. The values for each key are lists of candidate values. These are the hyperparameter ranges: (a) Unit ranges are 32, 64, 128, and 256.

(b) Batch size ranges are 8, 16, 32, 64, and 500.

(c) dropout rate ranges are 0.1, 0.2, and 0.3.

2. Create a Keras classifier using the create model function, with epochs set to 10 and the input shape formed according to the shape of the input data.

3. Use GA to perform hyperparameter tuning using the Keras classifier. Use the following parameters: estimator, param distributions, generations, population size, cv, and verbose.

4. Fit the best model returned by GA on the training data, with batch size set to the best batch size parameter found by GA. Also, evaluate the model on the test data.

Please note that the algorithm assumes the availability of training and test datasets, and it uses these datasets to evaluate the performance of the LSTM model with the given hyperparameters.

Evaluation setup

Data division is into training, validation, and testing sets. Table 4 shows the dataset setting for each repeated process. Each algorithm is first analyzed for the abrupt drift and then for the gradual drift. Base Learner and drift detector are two different things. Base Learner is a model that predicts whose results help detect drift. Naive Basie is used as a base learner due to its simplicity to focus more on the drift detector’s comparison. Three major categories for all drift detectors are two signals, i.e., drift warning and detection. Drift detection is the actual occurrence of drift, while drift warning indicates future drift occurrences. Massive Online Analysis (MOA) is a drift analysis tool used to repeat the experimentation with the same parameters to replicate the results (Gonçalves et al., 2014). The same model applies to the cloud datasets to analyze drift in the cloud.

Table 4 Dataset configurations.

Dataset	Instancesize	t	d	
Sine	40,000	10	100	
Stagger	40,000	1	100	
Mixed	40,000	40	40	
Hyperplane	40,000	100	100	
Google usage trace	60,000	1	2,500	

The techniques mentioned above compare in terms of miss detection rates and false alarms (Evaluation Time and Mahalanobis distance). Model configuration is set as follows based on the optimum results. The batch size is 64 for the best steps rather than smaller or larger gradient steps. Also, two hidden layers in the architecture are enough to deal with the complexity of the data. Thus, more layers to the model can increase the complexity and overfit the model. Adding a dropout layer in the LSTMDD optimizes the training and validation accuracy.

Results and discussion

Evaluation criteria

This section covers the results of the experiments concerning the evaluation parameters and prediction error. Prediction Error is the number of drifts detected when no drift exists. The evaluation criteria for the LSTM drift classification using all measures can be expressed as follows. Let TP represent the true positives, TN represent the true negatives, FP represents the false positives, and FN represent the false negatives. The following measures can then be defined to gain a better understanding of the correct drift detection: Accuracy measures the overall correctness of the model’s predictions.

(15) TP+TNTP+TN+FP+FN

Precision measures the proportion of true positives over the total number of positive predictions.

(16) Precision=TPTP+FP

Recall measures the proportion of true positives over the total number of actual drift instances.

(17) Recall=TPTP+FN

The F-score measures a model’s precision and recall, giving an overall measure of its performance.

(18) F−score=2⋅Precision⋅RecallPrecision+Recall

The false positive rate (FPR) measures the proportion of instances misclassified as drift even when they aren’t.

(19) FPR=FPFP+TN

False negative rate (FNR) measures how many actual drift instances are incorrectly classified as non-drift instances.

(20) FNR=FNTP+FN

Result analysis

These experiments are repeated with the same configuration, and results are summarized in the remaining Table 5. Each drift detector is executed to observe its performance on each sudden and gradual dataset. DDM is renowned in research for the detection of sudden drift. Abrupt drift occurs suddenly and changes the data in a short period. In experimentation, it has also performed well in sudden and gradual drift cases. In the case of abrupt and gradual drift, EDDM performed the worst for all the datasets, although it tends to perform well for gradual drift. Distance between the classification error means.

Table 5 Drift detectors average error.

Dataset	DDM	EDDM	ECDD	STEPD	Adwin	HDDMA	HDDMW	LSTMDD	
Sine	0.016	0.047	0.031	0.0312	0.0156	0.251	0.282	0.106	
Stagger	0.718	4.583	0.721	0.616	0.721	0.746	0.737	0.086	
Mixed	0.499	1.655	0.498	0.464	0.498	0.498	0.500	0.680	
Hyper	0.249	0.509	0.249	0.509	0.249	0.249	0.252	0.391	
Google	0.007	0.006	0.007	0.006	0.008	0.008	0.008	0.005	

Configurations

The algorithms from the Machine Learning for Data Streams (MOA) tool have used the same optimized parameters settings provided in the existing article (Gonçalves et al., 2014) and summarized in Table 6. A minimum number of instances allowed before the drift or change is detected is represented by “N”. A minimum number of instances not correctly classified before the drift warning is “ e”. Standard deviations that will be used to warn are referred to as “W”. Similarly, other abbreviations are the number of standard deviations to detect drift as “D” & “S”, “ W1” is the window size, max global error as “G”, and frequency of window reduction is as “M”.

Table 6 Drift detector parameters configuration.

Drift detector	N	D	W	Other parameters	
DDM	30	2.5	2	–	
EDDM	30	0.9	0.95	e =15	
ECDD	30	–	0.5	a = 40, l = 0.1	
STEPD	20	0.03	–	m1 = 0.08	
ADWIN	–	–	–	g = 0.02, M = 16	

Results

Table 7 summarizes results for abrupt drift datasets. In sine datasets, except EDDM, HDDMA, and HBBMW, all others have shown promising results. All others showed impressive results for sudden drift in the second dataset except for EDDM. ADWIN compares the two subsequent windows, and the proper selection of windows improves the results. It performed well for both by showing performance near DDM, but its performance depends upon the position of drift and window sizes. LSTM-Drift detector could detect drift well in case of both sudden and gradual datasets.

Table 7 Drift detector summary for abrupt drift.

Drift type	Abrupt	
	Average error	Dataset	Average time	Dataset	
DDM	Low	All	Low	Stagger	
ADWIN	Low	Sine	High	Stagger	
EDDM	High	ALL	High	All	
HDDMA & HDDMW	High	Sine	Low	All	
LSTMDD	Low	ALL	Low	All	

Drift detector performance on the gradual drift is summarized in Table 8. DDM was again able to give good results for gradual drift for both gradual datasets. The remaining drift detectors, including LSTMDD, performed nearly the same in gradual datasets. In Google usage trace, sudden and gradual drift occurred, and LSTMDD performed better than other datasets. LSTM networks are well-equipped to handle both short and gradual drift in datasets. When confronted with fast drift instances with a sudden and brief change in the CPU usage pattern, LSTMs quickly adapt to these changes due to their ability to retain crucial information over time. The memory cells within LSTM units enable the network to adjust its internal state and capture the new patterns associated with the drift, making them adept at detecting and responding to short-lived changes. For gradual drift, where the CPU usage pattern changes gradually over an extended period, LSTMs are also effective. By leveraging their capacity to capture long-term dependencies and remember past patterns, LSTMs can recognize the subtle shifts that occur over time and adapt to the evolving trends associated with gradual drift. By training on historical CPU usage data that encompasses instances of drift, LSTM networks can learn to identify and accurately classify short and gradual drift instances by modeling the temporal dynamics of the CPU usage sequence. Fine-tuning hyperparameters and optimizing the model architecture further enhance the LSTM’s performance in detecting drift.

Table 8 Drift detector summary for gradual drift.

Drift Type	Gradual	
	Average error	Dataset	Average time	Dataset	
DDM	Low	All	Low	All	
Other DD	Average	All	Average	All	
EDDM	High	All	High	Mixed	
LSTMDD	Low	ALL	Low	All	

Integrating an attention mechanism in an LSTM-based architecture enhances the model’s ability to handle imbalanced datasets. By allocating attention weights to specific parts of the input sequence, the model can focus on informative instances, particularly those belonging to the minority class or representing abnormal behavior in the CPU usage data. Thus, this helps mitigate the effects of class imbalance and enables the model to learn discriminating features and adapt to varying imbalance ratios. The attention mechanism provides a means to overcome the bias towards the majority class and improve the model’s performance in accurately detecting anomalies in imbalanced datasets.

The evaluation metric defined above to access the performance of LSTMDD against all databases is summarized in Table 9. Accuracy is not the best metric in the drift detection case, so precision and recall are the more critical evaluation metrics. Following the class balancing, each resource level’s confusion matrix is displayed in Fig. 6 for further explanation. Sampling was used to balance the classes because drift-occurring events are infrequent. Similarly, when determining the percentage of true drift cases mistakenly categorized as non-drift instances, FNR matters more than FPR. Low FNR is shown by the model in Table 9 for each of the three resource types. LSTMDD performed well on CPU usage drift detection with a macro f-score 0.98.

Table 9 LSTMDD model’s results on datasets.

Dataset	Accuracy	Precision	Recall	F-score	FPR	FNR	
Sine	0.85	0.85	0.85	0.85	0.15	0.13	
Stagger	0.76	0.78	0.76	0.75	0.39	0.09	
Mixed	0	0	0	0	0	0	
Hyper	86	0.86	0.86	0.85	0.09	0.18	
CPU google	0.98	1	0.98	0.98	0.019	0.0004	
Memory google	0.98	0.97	0.99	0.98	0.028	0.003	
Disk google	0.98	0.98	0.99	0.98	0.011	0.0001	

Figure 6 Confusion matrix of CPU, memory & disk.

Conclusion

Cloud usage constantly changes and requires an adaptive model because a static model, even trained with over a hundred years of data, will not be enough. Different drift detectors for adaptive learning mechanisms are required to deal with these sudden and gradual drifts. We presented a summary of the execution of different types of detectors for synthetic and real cloud datasets. The concept of drift detector selection depends on the domain for better detection, so generalization is impossible. In this article, we have investigated using LSTM models for concept drift detection in the cloud domain. This approach has shown promise in various applications, but its effectiveness for detecting concept drift in cloud environments still needs clarification. This article presents an attention mechanism-based LSTM using a genetic hyper-tuning drift detector to detect sudden and gradual drifts in the cloud. LSTMs’ ability to retain crucial information while capturing both short-lived and gradual changes is adept at detecting and responding to drift in data. At the same time, integrating an attention mechanism enhances their performance in handling imbalanced datasets. The DDM, EDDM, ADWIN, STEPD, ECDD, HDDMA, HDDMW, SEED, SEQ 1 and 2, and RMMD drift detectors are compared with the proposed LSTMDD. It performed well for datasets of larger size, with real drifts, with an f-score of 98% for all resource types.

In the future, we will enhance this model with built-in adaptive learning for the cloud to achieve better efficiency. The ensemble is best known for providing the advantage of multiple algorithms according to the requirement. The researcher can further explore the ensemble of drift detectors to improve the performance. We are exploring the potential use of the proposed approach for other applications beyond concept drift detection in cloud computing. For example, it may be possible to adapt the approach in other distributed computing systems where concept drift is a concern.

The authors acknowledge the CPInS Lab for providing hardware resources for experimentation and data verification and Artificial Intelligence and Data Analytics (AIDA) Lab at Prince Sultan University for data verification.

Additional Information and Declarations

Competing Interests

Author Contributions

Data Availability

The authors declare that they have no competing interests.

Tajwar Mehmood conceived and designed the experiments, performed the experiments, analyzed the data, performed the computation work, prepared figures and/or tables, authored or reviewed drafts of the article, and approved the final draft.

Seemab Latif conceived and designed the experiments, performed the experiments, performed the computation work, prepared figures and/or tables, authored or reviewed drafts of the article, and approved the final draft.

Nor Shahida Mohd Jamail analyzed the data, authored or reviewed drafts of the article, and approved the final draft.

Asad Malik conceived and designed the experiments, analyzed the data, performed the computation work, authored or reviewed drafts of the article, and approved the final draft.

Rabia Latif analyzed the data, authored or reviewed drafts of the article, and approved the final draft.

The following information was supplied regarding data availability:

The Google Trace Data is available at GitHub: https://github.com/google/cluster-data.

John Wilkes, Charles Reiss, Nan Deng, Md Ehtesam Haque, and Muhammad Tirmazi.

The synthetic datasets for concept drift detection purposes available at Dataverse Harvard: López Lobo, Jesús, 2020, "Synthetic datasets for concept drift detection purposes", https://doi.org/10.7910/DVN/5OWRGB, Harvard Dataverse.

The Synthetic Data Streams Repository is available at GitHub:

https://github.com/alipsgh/data-streams/tree/master/synthetic.

Pesaranghader, A., Viktor, H. & Paquet, E. Reservoir of diverse adaptive learners and stacking fast hoeffding drift detection methods for evolving data streams. Mach Learn 107, 1711–1743 (2018). https://doi.org/10.1007/s10994-018-5719-z.

This code is available at GitHub and Zenodo:

- https://github.com/Tajwarresearch/LSTMDD/.

- Mehmood, T. (2023). LSTMDD. In PeerJ Computer Science. Zenodo. https://doi.org/10.5281/zenodo.10203512.

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
