# Peer review of "LSTMDD: an optimized LSTM-based drift detector for concept drift in dynamic cloud computing"

_PeerJ Computer Science, doi:10.7717/peerj-cs.1827_

## Round 0.1 · original submission · Major Revisions

· Academic Editor

Major Revisions

Based on the reviewers’ comments, you may resubmit the revised manuscript for further consideration. Please address the reviewers’ comments carefully and submit a list of responses to the comments along with the revised manuscript.

**Language Note:** PeerJ staff have identified that the English language needs to be improved. When you prepare your next revision, please either (i) have a colleague who is proficient in English and familiar with the subject matter review your manuscript, or (ii) contact a professional editing service to review your manuscript. PeerJ can provide language editing services - you can contact us at [email protected] for pricing (be sure to provide your manuscript number and title). – PeerJ Staff

Reviewer 1 ·

Basic reporting

This study addresses concept drift in cloud computing, proposing LSTMDD as an effective solution for early detection and outperformance of other methods in detecting drift, leading to improved resource allocation and performance.

A. The last line of the abstract either merged with the first statement or should be removed to make it more appropriate
B. The reference style used is somewhat wrong e.g., it should be in parenthesis (Zaharia et al. 2011)
C. Line # 45, needs to be revisited. Currently, it does not make sense
D. Line # 56, The online learning model has benefits but it also has some costs which need to be discussed here
E. Line # 70, justification missing for using non-Gaussian distributed cloud computing environments
F. Overall, the introduction section does not strongly represent the problem statement with solid literature evidence. Secondly, the introduction section can be reduced while removing unnecessary basic domain information.
G. In the literature review section, domain knowledge is presented most of the time and in the last paragraph, the work done in this domain is discussed. The second part should be extended to support the problem statement
H. Table 3, if the dataset type is the same for all then it should be removed from Table 3
I. Consistency in the write-up is missing e.g., dataset, data set, data-set, Fscore, F-score
J. Figure 2 is used before Figure 1 in the text
K. Line # 273, In the Online learning mechanism, there are two phases, namely the Online and Offline phases, which 274 include a data distribution analyzer and a drift detector. How?
L. Heading 3.4 does not have any purpose here
M. All given tables and figures are not referenced in the text. Some are cited in the wrong sequence
N. The time unit missing in Figures 3, 4
O. The conclusion section should discuss something about results in a quantitative way
P. It will be better for the reader's point of view if related work is written in chronological order.
Q. Enough references have been cited.

Experimental design

Experimental design details are not provided, and results can not be reproduced because of the source code provision.

Validity of the findings

Different figures are given but how results are generated to draw these figures is missing.

Additional comments

Overall, a good article but some attention is required to the points mentioned above.

Reviewer 2 ·

Basic reporting

This paper mainly focused on classifying cloud workloads into three categories: sudden increase, gradual increase, and repeated workloads. Authors have used Naïve Basie model as a base learner and, in the next stage, used the basic LSTM model. Authors optimized the LSTM model hyperparameters by using a genetic algorithm. Finally, presented classification results by considering the CPU usage pattern of workloads.

A problem statement is clear, however, authors must address the following queries for more clarification regarding the proposed methodology and presented results. This paper needs major corrections to improve the paper more technically.

Experimental design

1. Researchers use more conventional/classical terms to define the problem statement and proposed methodology whenever they talk about cloud workloads. But, in this paper, authors have used non-technical terms to describe workloads and cloud environments. For example, cloud workload types: Abrupt, gradual, and reoccurring. Also in Page: 2, authors used “non-Gaussian distributed cloud computing environment” to describe a dynamic cloud computing environment. In the abstract, used like “using attention mechanism”. If you use these terms, others may think authors used AI tools to rephrase the sentences. Don’t use these kind of terms.

2. Page: 2 (84), in contribution, this sentence is wrong “This involves conducting an analysis of sudden and abrupt drifts, and evaluating the performance of each drift detector…".

3. As per the architecture (Fig. 2), authors mentioned about three parameters, they are CPU, RAM, Disk. But, considered only CPU parameter as an evaluation parameter.

4. Considered CPU parameter and classified the sudden and gradual workload. There is no clarification about how authors have mapped these two things, that gap is missing.

5. As per your architecture, the authors have used three models, they are Naïve Basie as a base learner, LSTM model, and the New model. There is no information about “New model” and what it does.

6. Drift detector will classify the workloads into two categories such as sudden and gradual. In the next step, for sudden workloads, new model will be assigned to train. Could you clarify more about this new model?

7. In formulas, subscripts are missing. For example, in pages 7, 9, 10, for LSTM formulas not used subscripts.

8. Page 10: (347) – the written statement is wrong, at first, it was mentioned that hyperparameters are tuned by Bayesian optimization and after that, it was mentioned as “Genetic algorithm hyperparameter tuning”. Clarification is missing.

9. What are the ranges you have considered for LSTM hyperparameters?

10. At first, it was mentioned as hyperparameters tuned by genetic algorithms, but, in page: 12 (377), fixed values have been considered for hyperparameters.

11. Figure 4. For evaluation criteria considered, two windows of workloads and compared in figure 4. However, this graph needs clarification regarding the given statement.

12. I recommend plotting the confusion matrix for your evaluation results along with tabular results.

Validity of the findings

More clarification is required and consider all the above points to improve the paper quality from a technical perspective.

---

## Round 0.2 · Minor Revisions

· Academic Editor

Minor Revisions

Please see the minor changes asked by one of the reviewers and submit the revised manuscript for further consideration.

Reviewer 1 ·

Basic reporting

Figure 1,2,3 & 6 should be improved.

Experimental design

ok

Validity of the findings

ok

Additional comments

All mentioned changes have been incorporated. Only figure 1 to 3 & 6 need to be improved.

Reviewer 2 ·

Basic reporting

The authors have addressed all the comments given by reviewers. I have gone through the other reviewer's comments. Now, this paper can be published with current content.

Experimental design

Architecture is clear and explained with algorithms.

Validity of the findings

Presented the results in the proper manner.

---

## Round 0.3 · accepted · Accept

· Academic Editor

Accept

Congratulations! The revisions are satisfactory and the manuscript is recommended for publication.